# CRISPR/Cas12-Based Ultra-Sensitive and Specific Point-of-Care Detection of HBV

**DOI:** 10.3390/ijms22094842

**Published:** 2021-05-03

**Authors:** Ronghua Ding, Jinzhao Long, Mingzhu Yuan, Xue Zheng, Yue Shen, Yuefei Jin, Haiyan Yang, Hao Li, Shuaiyin Chen, Guangcai Duan

**Affiliations:** 1College of Public Health, Zhengzhou University, Zhengzhou 450000, China; rhd23@foxmail.com (R.D.); LJZzzu@yeah.net (J.L.); 15660129935@163.com (M.Y.); zhengxue0313@163.com (X.Z.); shenyue5151@163.com (Y.S.); jyf201907@zzu.edu.cn (Y.J.); yhy@zzu.edu.cn (H.Y.); 2State Key Laboratory of Pathogen and Biosecurity, Beijing Institute of Microbiology and Epidemiology, Beijing 100071, China; lihao88663239@126.com; 3Key Laboratory of Molecular Medicine in Henan Province, Zhengzhou 450000, China

**Keywords:** Hepatitis B virus (HBV), CRISPR/Cas12a, LAMP, point-of-care detection

## Abstract

Hepatitis B remains a major global public health challenge, with particularly high prevalence in medically disadvantaged western Pacific and African regions. Although clinically available technologies for the qPCR detection of HBV are well established, research on point-of-care testing has not progressed substantially. The development of a rapid, accurate point-of-care test is essential for the prevention and control of hepatitis B in medically disadvantaged rural areas. The development of the CRISPR/Cas system in nucleic acid detection has allowed for pathogen point-of-care detection. Here, we developed a rapid and accurate point-of-care assay for HBV based on LAMP-Cas12a. It innovatively solves the problem of point-of-care testing in 10 min, particularly the problem of sample nucleic acid extraction. Based on LAMP-Cas12a, visualization of the assay results is presented by both a fluorescent readout and by lateral flow test strips. The lateral flow test strip technology can achieve results visible to the naked eye, while fluorescence readout can achieve real-time high-sensitivity detection. The fluorescent readout-based Cas12a assay can achieve HBV detection with a limit of detection of 1 copy/μL within 13 min, while the lateral flow test strip technique only takes 20 min. In the evaluation of 73 clinical samples, the sensitivity and specificity of both the fluorescence readout and lateral flow test strip method were 100%, and the results of the assay were fully comparable to qPCR. The LAMP-Cas12a-based HBV assay relies on minimal equipment to provide rapid, accurate test results and low costs, providing significant practical value for point-of-care HBV detection.

## 1. Introduction

Hepatitis B is an infectious disease caused mainly by liver lesions resulting from hepatitis B virus (HBV) infection, and is recognized as a global public health problem [1,2]. In 2015, the World Health Organization estimated that 257 million people worldwide have been infected with the HBV (HBsAg positive) and 887,000 died from chronic hepatitis B [3,4]. Approximately 68% of these infections were mainly recorded in developing countries in the medically poor western Pacific region and Africa [5,6]. The limited medical facilities contribute to the extremely low HBV detection rate in these areas. Therefore, HBV detection requires not only high sensitivity, but also point-of-care detection. At present, the clinical detection of HBV is mainly based on quantitative real-time PCR (qPCR) and serological testing. Although qPCR is highly sensitive, it requires specialized equipment and personnel, and is time-consuming, thus limiting its use in point-of-care testing [7,8]. HBV serological testing has been widely used in community hospitals due to its simplicity and rapid detection. However, its sensitivity, cross-reactivity, and specificity are poor. Additionally, it is less effective than nucleic acid testing, because the body does not produce the appropriate antigenic antibodies during the window period of infection [9,10,11]. Therefore, a rapid, accurate, low-cost, and convenient point-of-care testing technology for HBV testing should be developed. 

In recent years, clustered regularly interspaced short palindromic repeats (CRISPR)-associated (Cas), nuclease-based methods have provided a promising approach for rapidly adaptable, deployable detection [12,13]. Compared with other point-of-care assays, such as recombinant polymerase isothermal amplification (RPA) and loop-mediated isothermal amplification (LAMP), the CRISPR/Cas assay is more sensitive and specific [14]. Since the discovery of CRISPR/Cas detection technology, scientists have developed various detection platforms based on CRISPR/Cas systems, such as SHERLOCK (Cas13a), DETECTR (Cas12a), CDetection (Cas12b), and Cas14-DETECTR, which have successfully performed rapid, highly sensitive and accurate detection of various pathogens [15,16,17,18]. 

Here, we used LAMP amplification to address the limitations of poor amplification by RPA, the long PCR amplification time, and the need for specialized equipment for CRISPR/Cas detection [19]. Moreover, we addressed the challenge of sample nucleic acid extraction within 10 min by using rapid sample preprocessing. HBV DNA templates extracted from the sample preprocessing were amplified by LAMP, followed by mixing with the Cas12a–crRNA complex for detecting specific DNA target sequences. The presence of HBV DNA activates the collateral effect of Cas12a, then cleaves the fluorescent reporter release fluorescent signal. In addition, the combination of lateral flow test strip technology enables point-of-care test results to be visualized by the naked eye (Figure 1). This study is based on the LAMP method combined with the CRISPR/Cas12a detection system, which enables the detection of 1 copy/μL HBV DNA in a short time without relying on specialized equipment. The method of point-of-care testing for HBV provided in this study has important implications for the diagnosis and clinical management of hepatitis B in developing countries, where medical equipment is lacking.

## 2. Results

### 2.1. Construction of Cas12a-DETECTR System for HBV Detection

For Cas12a-DETECTR to detect various HBV subtypes, all HBV polymerase coding region 9199 sequences were sequence aligned to screen out conserved target regions, and then two crRNAs were designed accordingly (Figure 2a, Appendix A). After the Cas12a assay with 10^7^ copies/μL standard plasmid, the results showed that crRNA1 was efficient. Therefore, crRNA1 was selected for the subsequent detection of HBV in Cas12a-DETECTR (Figure 2c). PrimerExplorer v5 software was used to design three LAMP primers. After amplification of the 10^7^ copies/μL of standard plasmid, followed by agarose gel electrophoresis experiments, the results showed that primer 3 amplification efficiency was the best for the subsequent LAMP-Cas12a assay (Figure 2b,d, Appendix A). 

### 2.2. Analysis of the LoD and Specificity of the Cas12a-DETECTR System

The limit of detection (LoD), specificity, and detection time were used to evaluate the newly established HBV-DETECTR assay. A series of gradient dilutions of HBV standard plasmids was amplified using LAMP for 30 min and detected using the Cas12a-DETECTR fluorescence readout. The Cas12a-DETECTR assay had a low LoD of 1 copy/μL, while the variation from the negative control could be distinguished in 2 min (Figure 3a,b). 

Because the LOD results showed that the Cas12a-DETECTR assay was particularly sensitive, we therefore further explored the LOD of the Cas12a-DETECTR assay in terms of time. The HBV plasmids of 1 copy/μL were amplified by LAMP for different times, and then the Cas12a-DETECTR assay was performed. The results showed that after 3 min of Cas12a reaction, a significant difference was observed in the fluorescence values between the LAMP amplification for 10 min and the negative control. Hence, 1 copy/μL of HBV DNA could be detected by LAMP-Cas12a fluorescence readout within 13 min (Figure 3c,d).

In addition, the fluorescence readout results of the Cas12a-DETECTR assay showed that the fluorescence curve of LAMP amplification at 15 min and afterward was consistent (Figure 3c,d). To ensure the stability of the test results, we performed LAMP amplification for 15 min as the subsequent Cas12a lateral flow test strip incubation time gradient test. The results of the lateral flow test strips showed that the Cas12a-DETECTR showed positive results after 5 min of incubation. Next, after 15 min of incubation, all reporter molecules in the Cas12a detection system were cleaved, showing strong positive results (Figure 3e).

Importantly, the specificity of the newly established HBV-Cas12a assay platform was tested. First, we performed a BLAST analysis of the target sequences in crRNA1 and found no similar sequences except for each subtype of HBV. However, the sequence alignment with other hepatitis viruses (including HAV, HCV, HDV and HEV) revealed significant differences (Figure 3h). Next, we used Cas12a-DETECTR to perform identification assays for various hepatitis viruses. The assay results showed that Cas12a-DETECTR could specifically identify only HBV (Figure 3f,g).

### 2.3. Validation of DETECTR-Based HBV Assay for Clinical Samples

To assess the effectiveness of the HBV-DETECTR assay system in clinical samples, 73 serum samples were assayed using qPCR and Cas12a-DETECTR (Appendix A). The serum samples for the Cas12a-DETECTR assay were used for pre-processing using ultra rapid nucleic acid releaser. To improve the stability and sensitivity of the Cas12a assay platform in clinical validation, we adjusted and optimized the Cas12a-DETECTR system (Appendix A). Clinical samples were assayed using the adjusted Cas12a-DETECTR assay system by using fluorescent readout and lateral flow strip techniques. The results show that HBV detection based on the Cas12a-DETECTR’s fluorescence readout and lateral flow test strip method was consistent with the qPCR assay (Figure 4). The sensitivity, specificity, PPA, NPA, Youden’s index, and the ROC curve results showed an area under the ROC curve of 1, indicating that DETECTR-based HBV detection was fully comparable to qPCR (Table 1, Appendix A). Moreover, the Cas12a-DETECTR assay has advantages in terms of sample processing, assay time, and assay difficulty, and is fully capable of point-of-care testing of HBV samples (Table 2).

## 3. Discussion

HBV infection left without timely diagnosis and treatment may be transmitted to others and may develop into chronic hepatitis B, which can cause cirrhosis and hepatocellular carcinoma [20]. In the prevention and control of hepatitis B transmission, the most suitable detection technology is still being studied [21]. Importantly, qPCR and ELISA assays may not be excellent choices for HBV testing in developing countries in the western Pacific and African regions, where the prevalence of hepatitis B is high (Table 2) [22,23]. Therefore, rapid, accurate, and economical diagnostic tests that can be deployed at point-of-care are needed to prevent and control hepatitis B transmission. 

With the development of CRISPR/Cas-based systems for nucleic acid detection, rapid, accurate, and portable point-of-care detection of pathogens has become a reality. Although the highly sensitive detection of HBV based on PCR-Cas13a has been achieved, it still requires complex nucleic acid extraction of clinical samples, specialized PCR amplification, and fluorescence collection of Cas13a by the qPCR system, thus limiting the efficiency of detection and point-of-care testing [24]. 

Here, we developed the LAMP-Cas12a method for HBV detection to develop HBV detection. We used a rapid clinical sample processing method that allows for the extraction of nucleic acids in a short time frame, without the use of complex kits and specialized equipment, thus allowing for point-of-care testing. In the present study, we used the LAMP-based amplification method to eliminate the defects of previous CRISPR assays, which were its poor RPA amplification and the need for specialized equipment and time-consuming PCR amplification, while ensuring amplification efficiency [19]. More importantly, in combination with lateral flow test strip technology, the results can be visible to the naked eye, and have great practical value in home testing. Fluorescence readout can be achieved with portable fluorescence collection instruments, or with point-of-care detection by colorimetric analysis [25,26,27]. In the present study, the LoD of LAMP-Cas12a was 1 copy/μL. In addition, fluorescent readouts could be detected as early as 13 min and lateral flow test strips could be used for successful detection within 20 minutes (Figure 3a–e). More importantly, the LoD is comparable to that of qPCR, which is more advantageous in terms of efficiency and cost of detection, and does not rely on specialized equipment. However, the results were not ideal in pre-experiments of clinical sample testing. Therefore, the effect of the amount and purity of nucleic acids extracted, caused by the rapid sample pre-processing, was minimized by extending the LAMP incubation time to 30 min. In addition, the Cas12a-DETECTR assay system was optimized (Appendix A). After continuous optimization of the LAMP-Cas12a assay system and conditions, the sensitivity and specificity of the LAMP-Cas12a-based assay reached 100% for both fluorescent readouts and lateral flow test strips after validation evaluation of 73 clinical samples (Table 1, Figure 4). In addition, the method was found to be highly specific and resistant to interference (Figure 3e–g). Therefore, this study provides a LAMP-Cas12a-based HBV point-of-care assay that has high practical value in the future prevention and control of hepatitis B. 

## 4. Materials and Methods

### 4.1. Nucleic Acid Preparations

All HBV polymerase coding region sequences were obtained from the HBV database (https://hbvdb.lyon.inserm.fr/HBVdb/) and sequence alignment was performed using Clustal X software [28]. After sequence alignment, conserved sequences matching the Cas12a protospacer adjacent motif (5’-TTTN) were screened, and the corresponding CRISPR RNAs (crRNAs) were designed as guide RNAs (gRNAs) for subsequent HBV Cas12a assays [29,30]. Different crRNAs were performed by Cas12a-DETECTR for screening the optimal crRNA. Based on the screened optimal crRNA, the corresponding LAMP primers were designed using the PrimerExplorer v5 software (https://primerexplorer.jp/e/ accessed on 29 April 2021) (Eiken Chemical Co. LTD, Tokyo, Japan). BLAST and sequence alignment were used to analyze the specificity of the designed LAMP primers and candidate target sequences for the HBV genome. Moreover, crRNAs were designed, synthesized, transcribed, and purified as previously described [16,31] (Appendix A).

### 4.2. Loop-Mediated Isothermal Amplification (LAMP)

LAMP primers were synthesized by Shanghai Sangon Biotech. The LAMP reaction system was operated strictly according to the instructions (Sangon Biotech, Shanghai, China), and the amplification was carried out at 65 °C for 15–30 min (Appendix A). In addition, the amplified LAMP product was centrifuged and then placed on ice to prevent residual aerosol contamination on the EP tube lid.

### 4.3. HBV DNA Detection Based on Cas12a-DETECTR

The DETECTR assay was performed via steps. First, HBV DNA was pre-amplified using LAMP to generate more target sequence substrates for the Cas12a-DETECTR. Next, the detection of the amplified HBV DNA by Cas12a-DETECTR triggered the collateral cleavage of reporter molecules for lateral-flow or fluorescence assays, and this process was performed as previously described (Appendix A) [15]. For Cas12a-DETECTR fluorescence readout, the ABI fast 7500 was used to collect in real-time the fluorescence generated by the Cas12a cleavage of the fluorescent reporter (5′-/6-FAM/TTTTTT/BHO/-3′). The lateral flow test strip method enables visual field detection and is based on the principle that after Cas12a is activated to cleave the reporter (5′-/6-FITC/TTTTTTT/Biotin/-3′), the biotin end is captured in the first line (control line) and 6-FITC produces a positive band visible to the naked eye in the second line (test line) after binding to the anti-FITC and gold particles. In addition, the lateral flow test strip method requires incubation in a metal bath at 37 °C for 15–30 min before performing the test [32] (Figure 1).

### 4.4. Clinical Sample Collection and DNA Extraction

Clinical serum samples were provided by the First Affiliated Hospital of Zhengzhou University. Serum samples were collected by drawing 2 mL of venous blood from the subject with a sterile syringe, which was then injected into a sterile collection tube, followed by direct centrifugation at 1600 rpm for 5 min at room temperature to separate the serum. Finally, the sample was transferred into a 1.5 mL sterilized centrifuge tube for backup. Rapid HBV nucleic acid extraction was performed using ultra-rapid nucleic acid releaser (Appendix A). According to the instructions, 5 μL of nucleic acid releaser was added into a 1.5 mL centrifuge tube. Then, 20μL of HBV serum was added, and the solution was mixed gently. Then, the sample was placed in a constant-temperature metal bath, at 95 °C, for 5 min. Finally, after equilibration at room temperature for 2 min, the supernatant was centrifuged at 10,000 rpm for 2 min. Exactly 5 μL of supernatant was aspirated for LAMP amplification. To avoid the influence of subjective factors, we subjected the clinical samples in a blinded test by using the Cas12a-DETECTR assay (both the fluorescent readout and the lateral flow test strip method) and qPCR, respectively. Clinical samples were evaluated using qPCR as the gold standard, which is commonly used in clinical practice. In addition, the sensitivity, specificity, positive predictive value (PPV), negative predictive value (NPV), Youden’s index and receiver operating characteristic curve (ROC) were used to evaluate the effectiveness of the Cas12a-DETECTR assay. For the definition of positive samples for fluorescence readout, we set the signal-to-noise ratio parameter (the ratio of the fluorescence value of the sample to the negative control, S/N) to S/N>3 after 30 min of Cas12a reaction, which was considered an HBV-positive sample.

### 4.5. Statistical Analysis

The Cas12a-DETECTR assay was performed in three parallel experiments to avoid experimental error. Data were analyzed and graphed using SPSS 21 and GraphPad Prism 8.3.0 (GraphPad, Inc., La Jolla, CA, USA).

## Figures and Tables

**Figure 1 ijms-22-04842-f001:**
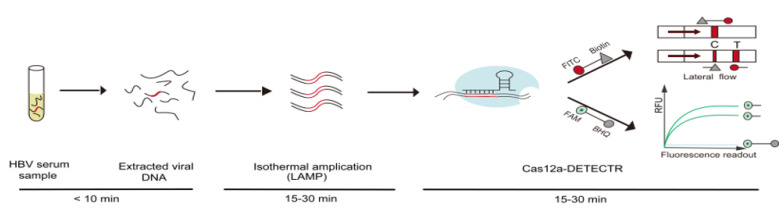
Cas12a-DETECTR assay of HBV DNA. After HBV serum samples were rapidly processed within 10 min, the extracted HBV DNA was amplified by performing LAMP of the target sequence (15–30 min). Cognate binding of the Cas12a–crRNA complex to amplified HBV DNA targets triggered the collateral activity of Cas12a, which cleaved ssDNA reporters (15–30 min). The cleaved ssDNA is visualized by both fluorescence readout (FAM-BHQ) and lateral flow test strips (FITC-Biotin).

**Figure 2 ijms-22-04842-f002:**
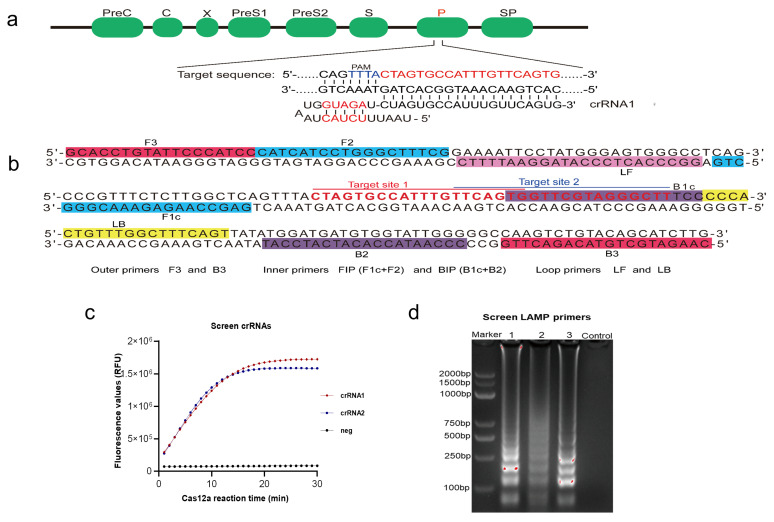
Preparation of Cas12a-DETECTR system. (**a**) HBV genome map shows target and crRNA sequences on the polymerase coding region. (**b**) The sequence and location of LAMP primers and target sites in the HBV genome. (**c**) To screen good crRNA, we performed Cas12a assays on 10^7^ copies/μL of HBV plasmids by using crRNA 1 and 2; neg: parallel control without crRNA addition for the same Cas12a assay. (**d**) To screen the best LAMP primer, after the amplification of different LAMP primers, we performed agarose gel electrophoresis.

**Figure 3 ijms-22-04842-f003:**
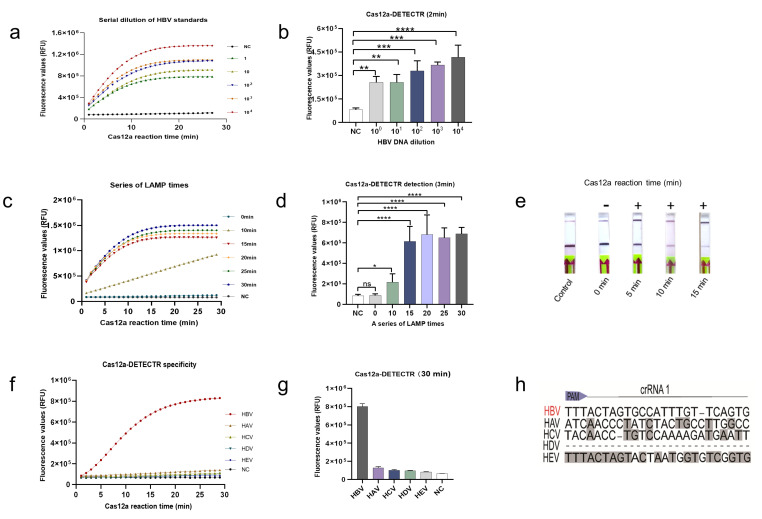
Determination of LoD and specificity of Cas12a-DETECTR assay. (**a**) Fluorescence curves generated by the Cas12a-DETECTR reaction at each dilution. Data were expressed as mean ± SD. from triplicate assays. Negative control (NC) utilized RNase-free water as input instead of HBV DNA dilutions. (**b**) Comparison of fluorescence values generated after 2 min of Cas12a-DETECTR reaction at each dilution. (**c**) Fluorescence curves generated by the Cas12a-DETECTR reaction at each LAMP incubation time gradient (HBV concentration is 1 copy/μL). (**d**) Comparison of fluorescence values generated after 3 min of Cas12a-DETECTR reaction at each LAMP incubation time gradient (HBV concentration, 1 copy/μL). (**e**) Comparison of lateral flow test strip results at each Cas12a-DETECTR reaction time gradient after 15 min of LAMP incubation (HBV concentration is 1 copy/μL). (**f**) HBV Cas12a-DETECTR fluorescence readout assays for different hepatitis virus clinical samples. (**g**) The fluorescence values generated after 30 min of Cas12a-DETECTR reaction for each virus. (**h**) The sequence alignment of nucleotides in the crRNA 1 target region of the HBV gene with other common human hepatitis viruses, including HAV, HCV, HDV, and HEV. +: indicates a positive sample, -: indicates a negative sample. **: *p* < 0.01; ***: *p* < 0.001; ****: *p* < 0.0001.

**Figure 4 ijms-22-04842-f004:**
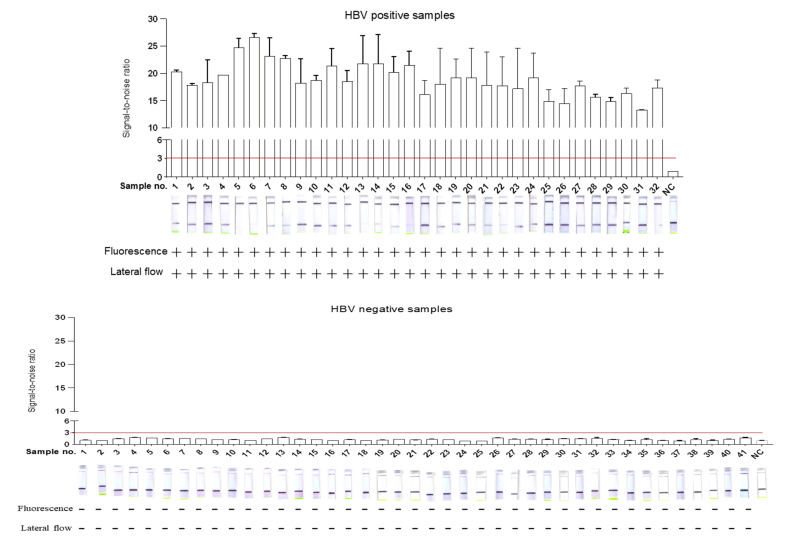
Cas12a-DETECTR assay for HBV in 73 clinical samples, which were tested using qPCR and Cas12a-DETECTR (including fluorescence readout and lateral flow test strip readout). Evaluation of Cas12a-DETECTR assay results with qPCR assay results. HBV positivity was defined based on qPCR results. +: indicates a positive sample, -: indicates a negative sample.

**Table 1 ijms-22-04842-t001:** Clinical validation of a DETECTR-based assay for the detection of HBV.

		qPCR		Sensitivity	Specificity	PPA	NPA	YI
		Positive	Negative	(95% CI)	(95% CI)	(95% CI)	(95% CI)	(95% CI)
Cas12a-DETECTR fluorescence readout	Positive	32	0	100%	100%	100%	100%	100%
	Negative	0	41	(86.7–100%)	(89.3–100%)	(86.7–100%)	(89.3–100%)	(76–100%)
	Total	32	41					
Cas12a-DETECTR lateral-flow readout	Positive	32	0	100%	100%	100%	100%	100%
	Negative	0	41	(86.7–100%)	(89.3–100%)	(86.7–100%)	(89.3–100%)	(76–100%)
	Total	32	41					

73 samples were used for clinical validation of the DETECTR assay. CI, confidence interval; PPA, positive predictive agreement; NPA, negative predictive agreement; YI, Youden’s index; the gold standard for HBV is qPCR.

**Table 2 ijms-22-04842-t002:** Comparison of Ca12a-DETECTR assay with existing assays.

	qPCR ^a^	ELISA ^b^	PCR-Cas13a	LAMP-Cas12a
Assay type	Nucleic acid assay	Antigen and antibody assay	Nucleic acid assay	Nucleic acid assay
LoD	30 IU/mL	Poor sensitivity	1 copy/μL	1 copy/μL
Nucleic acid extraction	Yes	No	Yes	No
Equipment requirements	qPCR	Enzyme-labeled instrument	PCR, qPCR	Metal bath
Complex operations	Yes	Yes	Yes	No
Cross-reactivity	No	Yes	No	No
Assay cost	Expensive	Economic	Economic	Economic
Sample-to-result time (approximate)	4 h	1–2 h	2–3 h	60–70 min
point-of-care testing	No	No	No	Yes
Clinical Application	Quantitative HBV DNA assay	Assisted qualitative assay	Pending clinical validation	Pending clinical validation

^a^ Sansure Biotech’s Hepatitis B Virus Nucleic Acid Assay Kit (Prominence HBV DNA) for qPCR Detection Kit; ^b^ HBV serological test refer to Shanghai Rongsheng Bio’s Hepatitis B virus surface antibody diagnostic kit (enzyme-linked immunoassay).

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
