# Peer review of "CRISPR/Cas12-Based Ultra-Sensitive and Specific Point-of-Care Detection of HBV"

_ijms, 2021, doi:10.3390/ijms22094842_

Round 1

Reviewer 1 Report

I would recommend the following major and minor suggestions:

  1. Please mention and explain Figure 1 in the introduction rather just mentioning in the material and methods.
  2. Improve the quality of figure 2, 3 and 4. It is hard to interpret the results in current quality of figures.
  3. Please right the results section clearly. 

Minor points:

Line 23: Consume – takes

Line 35: recoder- recorded

Line 35, 36, 37, 38, 39: Please write these sentences in clear and concise manner, I suggest not to repeat the same information.

Line 43, 44: Please rewrite the sentence

Line 49, 50: please rewrite the sentence

Author Response

Point 1: Please mention and explain Figure 1 in the introduction rather just mentioning in the material and methods.

Response 1:

Thanks for your reviewer comments. We have explained in detail about figure 1 in the introduction.

Point 2: Improve the quality of figure 2, 3 and 4. It is hard to interpret the results in current quality of figures.

Response 2:

Thanks for your reviewer comments. We are also aware that the figures are not clear, so we have reworked them. The quality of the revised figures far exceeds the requirements of the magazine. If you think they are still unclear, we have added them to the supplementary material.

Point 3: Please right the results section clearly.

Thanks for your reviewer comments. We have reworked the results section to ensure that it is clear to read. If there are any shortcomings, we would appreciate your help in pointing them out.

Point 4: Minor points

Thanks for your reviewer comments. We have revised accordingly to your suggestions. If there are still deficiencies, please point them out.

Line 35, 36, 37, 38, 39: Approximately 68% of these infections were mainly recorded in developing countries in the medically poor Western Pacific region and Africa. The limited medical facilities contribute to the extremely low HBV detection rate in these areas. Therefore, HBV detection requires not only high sensitivity, but also point-of-care detection.

Line 43, 44: HBV serological testing has been widely used in community hospitals due to its simplicity and rapid detection.

Line 49, 50: In recent years, clustered regularly interspaced short palindromic repeats (CRISPR)-associated (Cas) nuclease based methods provide a promising approach for rapidly adaptable, deployable detection. Compared with other point-of-care assays, such as recombinant polymerase isothermal amplification (RPA) and loop-mediated isothermal amplification (LAMP), the CRISPR/Cas assay is more sensitive and specific.

Reviewer 2 Report

Development of POV methods for viral infection detection in this time has a great importance. The methods and results presented in this paper contribute to extend the technological offer not only regarding the Hepatitis B. Data and methods are well presented and exhaustive, controls are adequate and discussion and conclusions are consistent.

Author Response

Thanks for your reviewer comments.  Based on Reviewer 1's suggestions, we additionally made modest revisions to the article to ensure that it would be of publication quality. If you have any other suggestions please point them out, and thank you for reviewing this article. Finally, wish you good health, success in your work and happiness of your family. 

Round 2

Reviewer 1 Report

I suggest the following minor corrections:

Line 21: Real Time

Line 189-190: Delete ‘medically underdeveloped’

Author Response

Point 1:

Line 21: Real Time

Line 189-190: Delete ‘medically underdeveloped’

Response 1:

Thanks for your reviewer comments. We have revised accordingly to your suggestions. If there are still deficiencies, please point them out. Thank you very much for your careful review of this article. Finally, wish you good health, success in your work and happiness of your family.